# Developing a Compost Quality Index (CQI) Based on the Electrochemical Quantification of Cd (HA) Reactivity

**DOI:** 10.3390/molecules28031503

**Published:** 2023-02-03

**Authors:** Ana C. Silva, Pedro Rocha, Dulce Geraldo, Ana Cunha, Juan Antelo, José P. Pinheiro, Sarah Fiol, Fátima Bento

**Affiliations:** 1Department of Chemistry, Centre of Chemistry, Campus Gualtar, University of Minho, 4710-057 Braga, Portugal; 2CRETUS, Department of Physical Chemistry, University of Santiago de Compostela, 15782 Santiago de Compostela, Spain; 3Department of Biology, Centre of Molecular and Environmental Biology (CBMA), Campus de Gualtar, University of Minho, 4710-057 Braga, Portugal; 4CRETUS, Department of Soil Science and Agricultural Chemistry, University of Santiago de Compostela, 15782 Santiago de Compostela, Spain; 5Laboratoire Interdisciplinaire des Environnements Continentaux (LIEC), Université de Lorraine/ CNRS, UMR 7360, F54501 Vandoeuvre-lès-Nancy, France

**Keywords:** compost, humic substances, metal binding, cadmium, AGNES, quality index

## Abstract

The present work demonstrates the use of Cd^2+^ as a reactivity probe of the fulvic acids (FAs), humic acids (HAs) and dissolved organic matter (DOM) compost extracts. Significant differences were observed between the extracts, with the HA extract showing the highest reactivity. Comparing the different composts, the largest reactivity variation was again observed for HA then FA and finally DOM extracts. The Cd^2+^ binding extent was used to calculate the quality of composts and compared with a reference of uncomposted organic fertiliser (FLW), leading to the definition of an operational scale of compost quality. The parameter equivalent mass of fertiliser (*m_EF_*) was used for this scale sorted the seven composts from 0.353 to 1.09 kg FLW, for compost of sewage sludge (CSS) and vermicompost of domestic waste (CVDW), respectively. The significance of this parameter was verified through a correlation analysis between binding extent and the effect of compost application on lettuce crop growth in a field trial. The results demonstrate the potentiality of FA and HA extracts as markers of compost bioactivity and the use of Cd^2+^ as a reactivity probe.

## 1. Introduction

Compost is a crucial component of the agricultural circular economy and plays a vital role in sustainable agriculture by improving soil and reducing contamination caused by synthetic fertilisers. Composting is the most efficient way to convert organic wastes into valuable products. With the recent European directive on mandating the separation of organic waste and household waste, a significant increase in compost production in Europe is expected in the next decade [1]. However, this increase brings new challenges in terms of compost quality. 

To evaluate the quality of compost, technical data sheets can be used to access the differences between composts. These data sheets provide various characterization data such as electrical conductivity, humidity, carbon/ nitrogen (C/N) ratio, cation exchange capacity (CEC), organic matter (OM) content, and trace metal concentrations. However, the fertilising efficiency of compost is assessed by crop field experiments, where surprisingly do not use any of the physico-chemical compost parameters as quality indicators [2]. To address this issue, we conducted a thorough characterization of composts of different origin using various physico-chemical techniques (e.g., elemental analysis, Fourier transform infrared (FTIR), Thermogravimetric analysis (TGA), differential scanning calorimetry (DSC)) and by a lettuce crop field experiment [3]. The results of the lettuce growth were not correlated with any of the compost physico-chemical parameters, including the essential nutrients such as N, P, and K. This lack of correlation may be due to the complexity of the compost samples and the presence of a significant number of inert materials, such as lignin [3].

In the field of soil studies, when a direct soil analysis is inconclusive, one way to address the problem is to carry out some sort of organic matter extractions, usually according to the procedure recommended by the International Humic Substances Society (IHSS). Soil organic matter is extracted with a strong base under a N_2_ atmosphere, leaving an insoluble fraction named humin. The soluble fraction, is further separated by precipitation at pH 1 into the acid insoluble humic acids (HAs) and the fully soluble fulvic acids (FAs) [4]. Both FA and HA are nanometric colloids with a radius of approximately 1 nm and 2 to 10 nm, respectively. However, they have a tendency to form larger aggregates depending on solution conditions such as pH and ionic strength. A similar classification system is used in compost studies.

The HA/FA are chemically heterogeneous and contain a wide variety of different acidic functional groups, typically from carboxylic and phenolic families. As result, these humic colloidal particles also carry a significant amount of negative structural charge resulting from the (de)protonation of these groups.

The ability of HA/FA to bind proton and metal cations is used for their characterization as the extent of this process is strongly associated with the nature of the functional groups present and the structure of the molecular aggregates formed [5]. The binding ability is crucial to understanding their role as plant biostimulants at the farmland level, which is most likely correlated with humic matter reactivity [6,7]. Additionally, the fraction of the organic substances present in these extracts is highly mobile and may impact the transport, speciation, and bioavailability of trace metal in the environment [8]. 

To investigate organic matter reactivity in solution, one effective method is to study its interaction with divalent cations such as cadmium, lead, or copper [9]. Trace metal cations interaction with organic matter can be studied using stripping electroanalytical techniques at very low metal concentrations [10]. Amongst these techniques, the Absence of Gradients and Nernstian Equilibrium Stripping (AGNES) is used in this work due to its ability to quantify the free metal ions present in solution quite similarly to an ion selective electrode [11].

In recent years there has been a great debate in literature about the representativeness of these humic and fulvic acids as avatars of the soil organic matter [12]. Some authors have recommended milder extractions for instance with 0.1 M CaCl_2_ [13] or artificial rain [14], as they may be more representative of the available organic matter in soil. Therefore, in this work we propose to investigate the organic extract’s reactivity using Cd^2+^ ions as probes, mainly due to the large *pK* range (Cd^2+^/OM) and to the very low concentration of Cd^2+^ naturally present in the compost. We aim to compare traditional extractions with milder ones and investigate the correlation between the reactivity of the different extracts evaluated using Cd^2+^ as a probe and compost effect on lettuce growth in a crop field experiment to study the meaningfulness of a compost quality index based on a physical-chemical parameter directly linked to the reactivity of HA/FA.

## 2. Results and Discussion

### 2.1. Study of the Reactivity of the Extracts by Cd^2+^ Complexation

Results of Cd^2+^ by FA, HA, and DOM extracted from each compost are shown in Figure 1 along with results from a uncomposted fertiliser for comparison. The plots show the complexation ratio, *K* (expressed in (µmol/L)^−1^) as a function of total Cd^2+^ concentration, *c_MT_*, (expressed in µmol/L). The *K* values, calculated as the ratio of complexed Cd^2+^ (*c_ML_*) to the product of free Cd^2+^ (*c_M_*) and organic matter (*c_L_*), are related to the stability constant of metal complexes. The concentrations are expressed in terms of the abundance of deprotonated groups at pH 7.0, *Q_pH 7_._0_*, which were calculated from acid–base titrations. The values of *Q _pH 7.0_* from the extracts can be found in the Appendix A. The plots in Figure 1 exhibit exponential-like variations, which are typical of heterogeneous materials. The binding capacity is strongly dependent on the metal to ligand ratio and the relative abundance of the strongest binding groups (carboxylic groups) is highest in the lower concentration range of added Cd^2+^ [15]. The decrease in the average *K* value follows the increasing involvement of weaker binding groups higher concentrations of added Cd^2+^.

The results shown in Figure 1 indicate that, in general, HA extracts have a higher binding capacity than FA extract, which in turn have higher complexing ability than DOM. This trend is likely due to differences in the chemical structure of these substances. HA is typically composed of larger, more highly condensed and cross-linked molecular structures than FA, which contribute to its higher binding capacity. Additionally, DOM typically contains a broader range of molecular structures and functional groups than HA ad FA, allowing for diverse complexing ability. 

There are two exceptions to this trend in compost of urban waste (CUW) and compost of livestock waste (CLW) samples. In the case of CLW, binding capacity of FA is comparable to HA and both are lower than that of DOM, which can be attributed to the specific composition of the humic and non-humic substances present in these extracts. 

Figure 2 compares the binding curves from extracts of the same nature. Each plot show results from the different composts and one fertiliser. The absolute variation in *K* values for a given extract can be used to evaluate the heterogeneity in each extract. The HA extracts from CA display the highest heterogeneity (*K* values vary from 5.9 (µmol/L)^−1^ to 0.63 (µmol/L)^−1^), while the CVDW, CVA, CLW, and FLW exhibit the lowest heterogeneities (e.g., for CVDW the *K* values vary from 0.77 (µmol/L)^1^ to 0.43 (µmol/L^−1^). The heterogeneity of the FA extracts is lower than that of the HA extracts. The FA extracts from CUW, CA, and CLW have higher heterogeneity (*K* values vary approximately between 0.38 (µmol/L)^−1^ and 0.06 (µmol/L)^−1^. Regarding the heterogeneity of the DOM extracts, CLW displays the highest one (*K* values varying between 2.75 (µmol/L)^−1^ and 0.41 (µmol/L)^−1^) as opposed to CVA which displayed the least heterogeneous extract (*K* values vary between 0.03 (µmol/L)^−1^ and 0.01 (µmol/L)^−1^).

The relative position of the experimental points of each sample on the plots in Figure 2 provides information on the relative reactivity of each type of extract. The FA extracts show similar reactivity among this set of materials, as the data points of the different samples are very close to each other. The HA extracts from CA stands out with the highest values of *K* and the remaining samples can also be differentiated. The DOM extracts from CLW displayed the highest values followed by CUW, while the points from the remaining composts appear very close to each other.

### 2.2. Evaluation of the Extent of Cd^2+^ Binding by the Different Extracts

While the reactivity of the extracts is important for their characterization, its operational relevance is limited in compost applications, such as in agriculture. The effectiveness of the interaction between HA, FA, and DOM with the environment (either soil or plants) depends not only on the reactivity of the extracts but also on the content of each extract in the compost. To establish the appropriate compost dose for the desired effects, it is necessary to quantify the abundance of each extract which can be expressed in terms of the carbon content of each humic extract, *C_HS_*, and dissolved organic matter, *C_DOM_* (in g_C_ kg_compost_^−1^). 

The carbon content of the HA and FA extracts for each sample is compared with that of the DOM in Figure 3a. The data used to calculate the results in Figure 3a, including carbon content and extraction yields, are shown in Appendix A. For most of the characterized composts and the fertiliser, the amount of carbon of DOM is similar to that of HA, with a ratio *C_HA_*/*C_DOM_* ≈1. However, for samples CVA, CA, and CDDW, the HA extracts contain 8 to 4 times more carbon than the respective DOM. The FA extracts in all samples contain lower amounts of carbon than the respective DOM, with ratios of *C_FA_*/*C_DOM_* less than 1 for all composts and an average value of 0.35.

Figure 3b compares the abundance of acid sites in humic extracts (*M_T,HS_^S^*) and DOM (*M_T,DOM_*). The acid sites abundance, expressed in mol kg_compost_^−1^, was calculated using the proton titration values *M_T_* (mmol g_C_^−1^) from Appendix A [16] and the carbon content (expressed in g_C_ kg_compost_^−1^) of each extract (*C_HS_* or *C_DOM_*).

The high correlation (r = 0.9) found between M*_T_* values of both FA and DOM extracts is likely due to their similar composition. The solubility of the FA in neutral or slightly acidic solutions means that it may be a significant fraction of the organic species present in DOM, along with other hydrophilic substances, such as transphilic acids, which are known to play a significant role in the fouling of ultrafiltration membranes in water treatment plants that are recovered (and can be recovered using a XAD-4 column) [17,18]. Additionally, the abundance of acid sites *per* kg of compost in DOM is roughly 20 times higher than that found for the FA extracts. This discrepancy may be due to the purification process used to obtain FA extracts, which may result in a loss of low molecular weight organic acids. Additionally, the hydrolysis of divalent (Ca, Mg) and trivalent cations (Al and Fe) present in DOM (with values ranging from 28.6 mg kg^−1^ (CA) to 171.1 mg kg^−1^ (CSS)) [3], may also contribute to the overestimation of the acid sites in DOM compared with the extensively washed FA extracts.

The average abundance of acid sites in HA was found to be 5 times lower than that of the DOM extracts when expressed in kg_compost_^−1^. This discrepancy may be due to the significant loss of carbon content in the humic fraction during purification and the presence of high levels of divalent and trivalent cations. In terms of carbon content and abundance of acid sites, DOM appears to be more significant than humic extracts when it comes the impact on soil acid-base properties. This is particularly relevant when solid compost is applied directly to farmland rather than using of HA/FA formulations that are commercially available or prepared by those with technical expertise. 

After determining the carbon content and the amount of acid groups, the comparison between the extracts was conducted by measuring the extent of the Cd^2+^ binding, referred to as *c_ML_*. This value, expressed in mmol kg_compost_^−1^ was calculated by referencing the Cd^2+^ binding by each extract to 1 kg of compost. The calculation was based in complexation data at pH 7.0 and took into account the abundance of the extracts using Equations (1) and (2) for the humic substances and water-soluble extracts, respectively: (1)$$c_{ML}\left(mmol\;kg_{compost}{}^{-1}\right)=\frac{c_{ML}\left(mol\;L^{-1}\right)}{c_{HS}\;\left(kg_{HS}L^{-1}\right)}\times Y\left(g_{HS}\;kg_{compost}{}^{-1}\right)$$
(2)$$c_{ML}\left(mmol\;kg_{compost}{}^{-1}\right)=\frac{c_{ML}\left(mol\;L^{-1}\right)}{\;F_{d}\times 50\;\left(g_{compost}L^{-1}\right)}\times 10^{6}$$
where *c_ML_* (mol L^−1^) and *c_HS_* (kg*_HS_* L^−1^) are the concentrations of metal complex and of organic matter (HA and FA), respectively, obtained from the electrochemical assay; Y is the yield of the HA/FA extractions (in g*_HS_* kg_compost_^−1^), *F_d_* is the dilution factor (*F_d_ = V_DOM_/V_Total_*) and 50 corresponds to the amount of compost *per* litre used to prepare the equilibrium solution (g*_compost_* L^−1^).

From Figure 4a, it is observed that the binding extent of the FA extracts for CA, CSS, CVA, and CDDW is similar to that of DOM, but for CVDW, CUW, and CLW it is lower than that of DOM. This is despite the fact that carbon content and acid sites abundance is much higher for DOM. The slope of the straight lines defined by the experimental points vary from about 1 (for CA, CSS, CVA, and CDDW) to 0.2 (for CUW and CLW), which is considerably higher than the corresponding *M_T_* ratio (0.022, obtained from the slope of the straight line in Figure 3b). This suggests that DOM is less reactive with Cd^2+^ compared with FA, indicating that the chemical structures present in the two extracts are likely different. A similar conclusion can be drawn from the comparison between the extent of binding of HA and DOM shown in Figure 4b. In this case, all the experimental points, except CLW, are above the *y = x* line, despite the carbon content of the majority of the HA extracts (CVDW, CLW, CSS, FLW, and CUW) being comparable to that of DOM and the acid sites abundance being 5 times lower on average than that of DOM.

### 2.3. Comparison of Extent of Cd^2+^ Binding of Compost with That of an Uncomposted Organic Fertiliser

The extent of Cd^2+^ binding of the extracts from different composts can be compared with that of identical extracts from an uncomposted livestock fertiliser (manure) which serves as a reference fertiliser. The results from the three types of extracts, FA, HA, and DOM are depicted in Figure 5. On the vertical axis the values for the composts are represented, and on the horizontal axis the values from the fertiliser are represented. The position of the experimental points of the different composts relative to the *y = x* line, which represents equivalence to FLW, provides a simple way to compare the binding extent of all compost extracts. The FA extract of CLW shows the most extensive binding of Cd^2+^ and is the only sample whose FA binds more extensively than the FA of FLW (data located above the *y = x* line) (Figure 5a). The FA extracts of all the remaining composts are below this line, indicating lower binding ability than FLW. For the HA extracts, CLW, CVDW, and CA, whose points are distributed along the *y = x* line (Figure 5b), have a Cd^2+^ binding extent comparable to FLW, whereas CSS, CDDW, CVA, and CUW, whose points are below the *y = x* line, have lower binding ability than FLW.

For the DOM extracts all points are well below the equivalence line (Figure 5c). CLW is the only sample that stands out from the remaining compost samples whose results are very close to the equivalence line. 

Based on these results, we suggest using Cd^2+^ as probe to compare different composts based on the extent of binding of this metal cation by the HA extract. This ranking highlights composts that contain higher amounts of humic acids, with greater abundance of acid groups and extensive Cd^2+^ binding. In other words, the extent of Cd^2+^ binding provides a measure that takes into account both the reactivity and the abundance of humic acids.

The HA extracts were selected for this evaluation due to their relative abundance, the larger absolute values of *c_ML_*, and larger differences observed between the *c_ML_* values of the different composts. Using this parameter, *c_ML_*obtained from the HA extracts, a ranking can be established by means of an operative scale based on the ratio (*c_ML_*)*_compos_*/(*c_ML_*)*_FLW_* that compares the reactivity of the HA extracts of composts with respect to the organic fertiliser (corresponding to the slopes of the regression lines in Figure 5b). This parameter, designated as equivalent mass of fertiliser (*m_EF_*)_,_ represents the mass of fertiliser required for binding an amount of Cd^2+^ as the HA extracts present in 1 kg of compost. This parameter may be used as a quality index to compare composts in terms of different features, such as the price, compared with the application of FLW. The calculated values of the parameter equivalent mass of FLW (*m_EF_*) are displayed in Figure 6a. In this ranking the composts CVDW, CA, and CLW (with *m_EF_*, values of 1.09, 1.04, and 0.906 kg of FLW, respectively) are in the top places, followed by CDDW and CUW (with 0.589 and 0.550 kg of FLW, respectively) and then CVA and CSS (with 0.409 and 0.353 kg of FLW, respectively). 

### 2.4. Verification of the Potential of the Reactivity Parameter, c_ML_, as a Marker of Compost Bioactivity

The proposed reactivity ranking, based on the binding extent of the extracts, can only be considered valid if there is a correlation between this parameter and the ability of the corresponding composts to promote crop productivity. We attempted to analyse this correlation using results from a previous study where we compared the agricultural effect of the composts CUW, CVA, CA, and the fertiliser FLW in a field study using lettuce (*Lactuca sativa* L.) as a model plant, evaluating the total leaf area (*TLA*) of the lettuce plants 5 weeks after planting [3]. The three compost samples and the fertiliser were applied in the field at the doses recommended by the producers. The treatment effect was also evaluated for two doses of CUW and FLW. Using the previously reported *TLA* values, a correlation analysis was performed with respect to the extent of the Cd^2+^ binding by each extract (calculated from the amount of extract that would be obtained considering the dose of compost/fertiliser placed in each bed) for an addition of 0.10 µmol L^−1^ and 0.30 µmol L^−1^ of Cd^2+^. The extent of binding for the lowest and highest concentrations of added Cd^2+^ are *c_ML,L_* and *c_ML,H_*, respectively. The obtained regression parameters are displayed in Table 1. While excellent correlations are obtained for HA and FA, for both *c_ML,L_* and *c_ML,H_*, indicating a strong association between the *TLA* and the extent of the Cd^2+^ binding, for DOM the correlation coefficient is almost zero. This result indicates that the species present in the DOM extract cannot be considered as markers of the compost/fertiliser bioactivity, in contrast to HA and FA. The proximity between the *TLA* value from the control assay (which was not included in the regression analysis) and the intercept of the regression lines support the validity of the correlations found for HA and FA to some extent. It is worth noting that the values for CVA were not included in these correlations due to their disagreement with the general trend defined from the results of the two remaining composts and the fertiliser, including two doses from two of the samples (CUW and FLW), in a total of five independent results. Despite the results of the HA and FA extracts being apparently similar in terms of the adjustment obtained, for HA there is a better proximity between the intercept of the correlation lines and the *TLA* value from the control assay. On the other hand, as the values of *c_ML,L_* and *c_ML,H_* of HA are higher it allows a better differentiation between composts in absolute terms, making it more suitable for the definition of a quality ranking. As the quality of the correlations obtained with *c_ML,L_* and *c_ML,H_* are similar, it can be concluded that *c_ML_* is robust with regard to the added Cd^2+^ concentration, despite the heterogeneity of acid sites that implies an increasing participation of weaker groups to increasing concentrations of added Cd^2+^. 

The plot in Figure 6b shows the correlation between the mean values of *TLA* and *c_ML,L_* for the HA, with the straight line representing the correlation line for five out of six independent results. For comparison, the plot for *c_ML,H_* can be found in the Appendix A.

Although the verification of the proposed parameter is limited, as it was performed with a reduced number of composts and a single culture, it supports further research on the use of this reactivity parameter in assessing composts quality.

The most surprising result of this study is the correlation between a result from a culture grown under natural field conditions (*TLA*) and a reactivity parameter purified humic extracts, which do not contain the fraction of the most soluble constituents.

The fact that the most effective markers of compost bioactivity are not the most accessible from the point of view of ease of release into soil through irrigation or precipitation may seem intriguing. However, several processes in the rhizosphere can facilitate the adsorption of bioactive components, such as root exudates rich in organic acids, which can promote the separation of humic aggregates and provide bioactive substances (phytohormones, HA, and FA) that can induce plant growth and other physiological, biochemical, and metabolic changes [7]. 

These organic acids can modify the size of humic substances aggregates [19], allowing for the penetration of the constituents of lower molecular weight into cells that can induce various responses in plants, such as the polar transport of auxin and stimulation of root plasma membrane H^+^-ATPase [20], as well as mediating the direct transport of nutrients [21]. These effects demonstrate the existence of intense soil–root–stem crosstalk mediated by HA/FA [22].

## 3. Materials and Methods

### 3.1. Identification of the Samples

The compost samples used in this study were produced from a diversity of raw materials and through different composting processes (e.g., industrial composting, domestic composting, and vermicomposting). The feedstock was predominantly of plant origin in the vermicompost of domestic waste (CVDW), in the domestic compost of domestic waste (CDDW) and in the compost of urban waste (CUW). Manure was incorporated in the vermicompost of algae (CVA), in the compost of livestock waste (CLW), in the compost of sewage sludge (CSS), in the sewage sludge in CSS and algae in CVA, and in the compost of algae (CA). Detailed information on raw material composting procedure is compiled in Appendix A. An organic fertiliser (100% animal waste, mainly chicken manure) commonly used in agriculture, was also included in the study for comparative purposes. The detailed compost characterization was previously reported in Silva et al. (2022) [3]. The C/N molecular ratios of all samples are displayed in Appendix A.

### 3.2. Preparation of the OM Extracts from Compost Samples

Three different extracts were obtained from each sample. The extractions and purification of the humic substances (HSs), fulvic-like acids (FAs) and humic-like acids (HAs) were carried out following the method recommended by IHSS [4]. Briefly, samples were extracted with 0.1 mol L^−1^ NaOH under an atmosphere of N_2_ at an extractant to compost ratio of 10:1 (*v*:*w*). The extracted HS were then separated into HA and FA fractions by acidifying the extract to pH 1 using a 6 mol L^−1^ HCl solution. The precipitate (HA) and the supernatant (FA) were separated by centrifuging at 2000 rpm for 20 min. The HA fraction was suspended in a solution of 0.1 mol L^−1^ HCl/0.3 mol L^−1^ HF to remove mineral impurities and then dialysed until the elimination of Cl^-^. The FA was purified by using an adsorption resin, XAD-8, and the alkaline eluate was passed through an H^+^-saturated cation exchange resin. The water-soluble extracts containing the dissolved organic matter (DOM) and water-soluble inorganic components that are also called equilibrium solutions were extracted using a less aggressive procedure. Succinctly, 2.50 g of the compost were placed in 50 mL of ultra-pure water at natural pH for 5 days in an open system. After 5 days (equilibration time), the supernatants were centrifuged (6000 rpm for 20 min) and isolated for characterization.

### 3.3. Acid–Base Titrations, Extraction Yields and Carbon Content of OM

The proton titrations of the different extracts, extraction yields and carbon content of the compost of algae (CA) and the uncomposted organic fertiliser (FLW) were carried out as described in López et al. (2021) [16]. The corresponding values from the extracts of all other composts used in this work were also presented in López at al. (2021) [16] and are compiled in the Appendix A for easy consultation.

### 3.4. Cd^2+^ Quantification by AGNES in OM Extracts for Reactivity Characterization

Solutions were prepared using ultrapure water (Milli-Q System, Millipore, Merck, Milwaukee, WI, USA). Cd^2+^ stock solutions were prepared by dilution of 1000 mg L^−1^ (Certipur, Merck, Milwaukee, WI, USA) Cd^2+^ standard solutions. The ionic strength of the solutions was adjusted to 12 mmol L^−1^ using 1.0 mol L^−1^ HNO_3_ (for analysis, ISO, PanReac Applichem, Darmstadt, Germany) and 1.0 mol L^−1^ NaNO_3_ (Suprapur, Merck, Milwaukee, WI, USA). The pH of the solutions was adjusted to pH 7.0 using 0.1 mol L^−1^ NaOH (Merc, Milwaukee, WI, USA).

An Autolab PGSTAT30 potentiostat connected to a Metrohm 663 VA Stands controlled by means of the GPES 4.9 software (EcoChemie, Metrohm, Herisau Switzerland) was used to carry out the electroanalytical measurements. The electrochemical cell consisted of a working glassy carbon rotating disk electrode (6.1204.300, Metrohm, Herisau Switzerland) modified by a thin mercury film, an auxiliary carbon rod electrode (6.1248.040, Metrohm, Herisau Switzerland) and a Ag/AgCl reference electrode (Driref-5, World Precision Instruments, Hertfordshire, UK). The thin mercury film electrode was prepared following a published procedure [15]. The pH measurements were carried out using a pH meter (CyberScan pH 510, Thermo Scientific, Rockford, IL, USA) connected to a glass combined electrode (HI-11311, Hanna instruments, Leighton Buzzard, UK).

AGNES was used to quantify free Cd^2+^ ions in the presence of the different FA, HA, and DOM extracts. Detailed instruction to perform AGNES experiments can be found in the excellent tutorial review by Solis et al. [23] including the experimental steps to implement AGNES, step-by-step laboratory protocols, and two practical case studies. In our case the methodology includes two steps: first a calibration step carried out at pH 4.0 and 12 mmol L^−1^ ionic strength were the blank and four aliquots of Cd^2+^ ions were measured. The second step consists of the Cd^2+^/OM titrations. At the end of the calibration a fixed amount (between 0.68 and 13.6 mg L^−1^) of the OM extract is added to the electrochemical cell and the pH is set to 7.0 by NaOH 0.1 mol L^−1^ addition. The total concentration of Cd^2+^ of the first titration point is the same as the last calibration point corrected by the dilution effect. The titration proceeds by several additions of known aliquots of a standard Cd^2+^ solution (10^−5^ or 10^−6^ mol L^−1^) so that the total Cd^2+^ concentration ranges between 10^−8^ and 10^−6^ mol L^−1^. All AGNES measurements were carried out in triplicate.

## 4. Conclusions

The reactivity of the DOM, FA, and HA extracts allow for differentiation between extracts of different origin, particularly the HA extracts whose reactivity varies more markedly. The interaction of extracts from different composts with Cd^2+^ was used to compare composts with each other and with an uncomposted fertiliser (FLW). The differentiation between composts was more apparent when this comparison was performed using HA extracts. A scale of compost quality is proposed, expressed in units of equivalent mass of fertiliser. This scale takes into account the abundance and reactivity of the HA extracts, using the experimental values of the metal complex concentration *c_ML_*. In this ranking, the composts characterized in this work were ordered as follows: CVDW > CA > CLW > CDDW > CUW > CVA > CSS. The validity of this scale was tested by comparing results from the reactivity parameter with growth results from a lettuce crop treated with some of these composts. The high correlation obtained between these two variables demonstrates that the proposed reactivity parameter, used to define the compost quality scale, has a high potential to be used as a marker of the bioactivity of composts.

Despite being the first time that a direct link between the bioactivity of the composts and their HA/FA properties has been demonstrated, the results of the present study can be seen as a direct consequence of the accumulated knowledge of the biological activities and plant responses induced by humic substances. For HA/FA extracts, these effects have been identified and related to different factors, such as dosage, origin, molecular size, degree of hydrophobicity, and aromaticity [19], and that the mode of production and the raw materials are relevant for the performance of the composts in the field.

The proposed methodology is adequate for the classification of composts, but additional experiments to validate on a large scale are needed to validate the use of *c_ML_* as a marker of compost bioactivity.

## Figures and Tables

**Figure 1 molecules-28-01503-f001:**
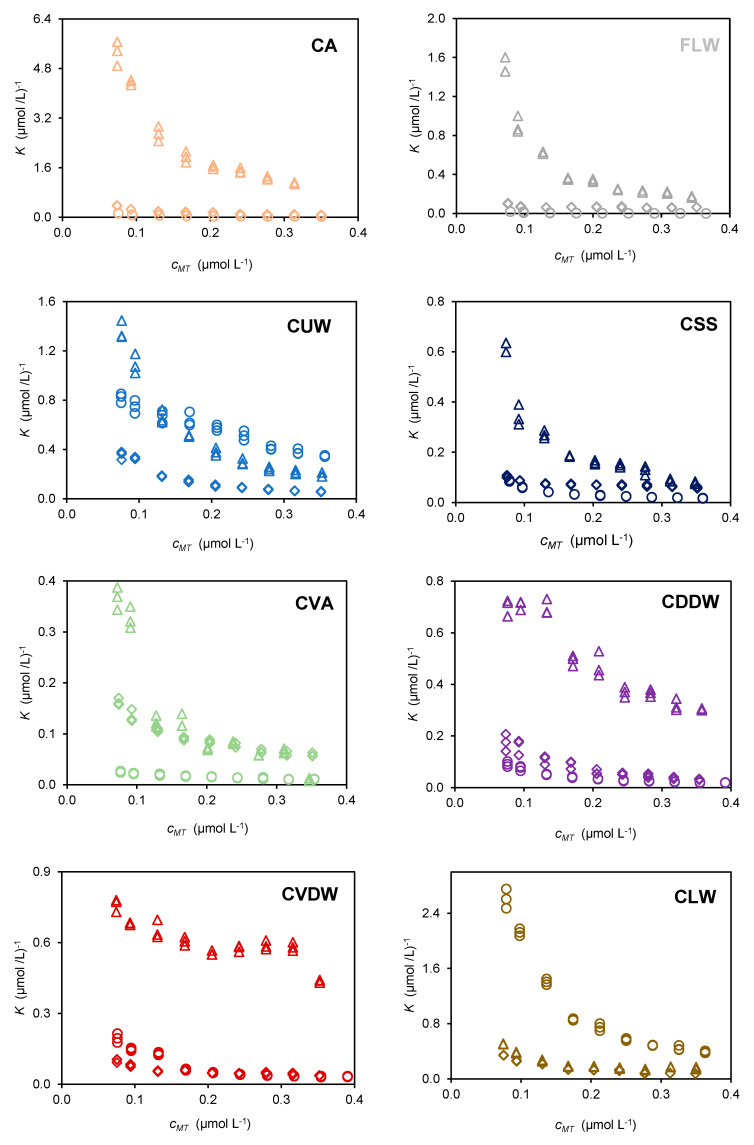
Experimental data expressed as complexation ratio, *K*, as a function of the total Cd^2+^ concentration, *c_MT_* at pH 7.0 and ionic strength 12 mmol L^−1^ for FA (◊), HA (∆), and DOM (o) extracted from 7 different composts (CA, CUW, CSS, CVA, CDDW, CVDW, and CLW) and one fertiliser (FLW).

**Figure 2 molecules-28-01503-f002:**
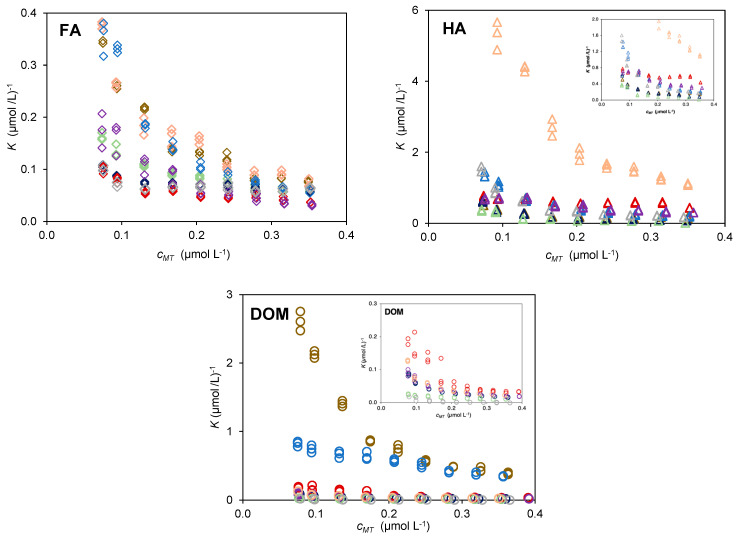
Experimental data expressed as complexation ratio, *K*, as a function of the total Cd^2+^ concentration, *c_MT_* at pH 7.0 and ionic strength 12 mmol L^−1^ in NaNO_3_ and HNO_3_ for FA, HA, and DOM extracted from: CDDW (purple), CVDW (red), CVA (green), CA (coral), CUW (blue), CSS (dark blue), CLW (brown), and of the fertiliser FLW (grey).

**Figure 3 molecules-28-01503-f003:**
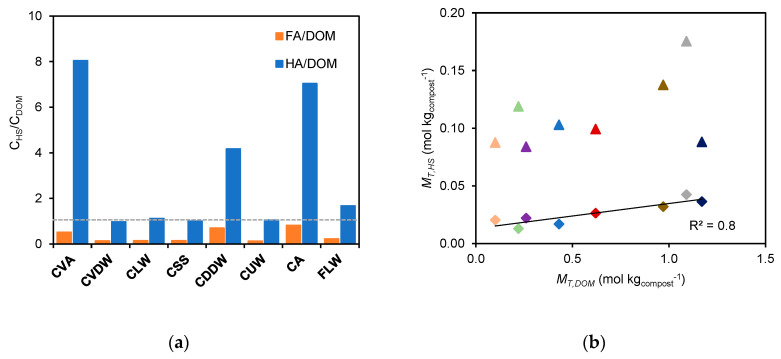
(**a**) Ratio between the carbon content in the humic extracts (FA and HA) with respect to the present in DOM (*C_HS_*/*C_DOM_*), the horizontal line is settled for *C_HS_*/*C_DOM_* = 1 and (**b**) correlation between the abundance of acid sites in FA (◊) and HA (∆) and the abundance of acid sites in DOM of the composts: CDDW (purple), CVDW (red), CVA (green), CA (coral), CUW (blue), CSS (dark blue), CLW (brown), and FLW (grey).

**Figure 4 molecules-28-01503-f004:**
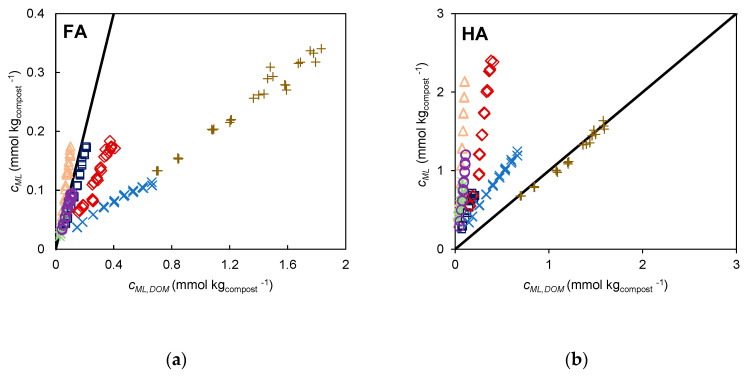
Comparison of Cd^2+^ binding data at pH 7.0 and ionic strength 12 mmol L^−1^ in NaNO_3_ and HNO_3_ for: (**a**) FA vs. DOM and (**b**) HA vs. DOM of composts: CDDW (purple (◯)), CVDW (red (◇)), CVA (green (∗)), CA (coral (△)), CUW (blue (✕)), CSS (dark blue (□)), CLW (brown (+)). The *y = x* line represents the equivalence between results.

**Figure 5 molecules-28-01503-f005:**
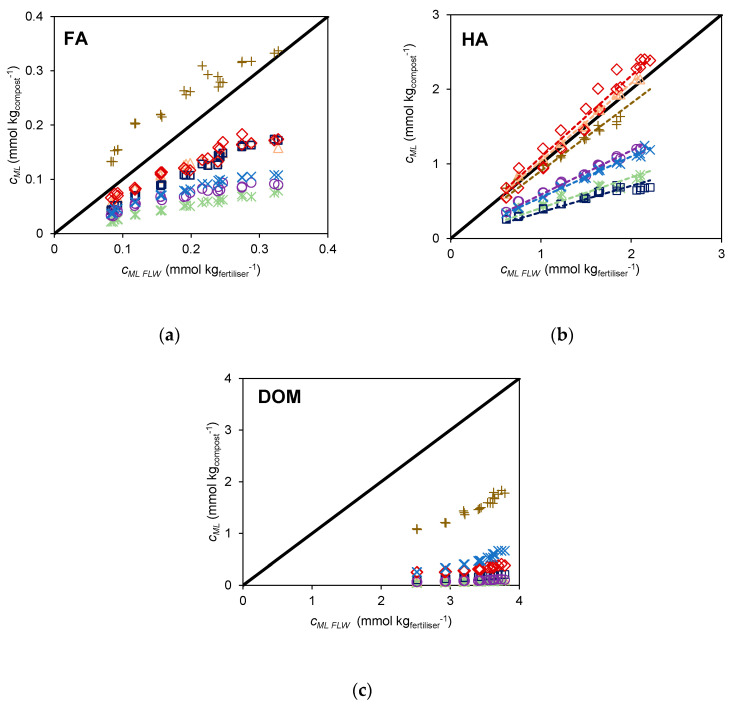
Comparison of Cd^2+^ binding data of extracts obtained by samples of different origin (CDDW (purple (◯)), CVDW (red (◇)), CVA (green (∗)), CA (coral (△)), CUW (blue (✕)), CSS (dark blue (□)), CLW (brown (+))) at pH 7.0 and 12 mmol L^−1^ ionic strength: (**a**) FA extracts vs. FA-FLW, (**b**) HA extracts vs. HA-FLW and (**c**) DOM extracts vs. DOM-FLW. The *y = x* line represents the equivalence between results. The dotted lines in (**b**) correspond to the linear fit correlations.

**Figure 6 molecules-28-01503-f006:**
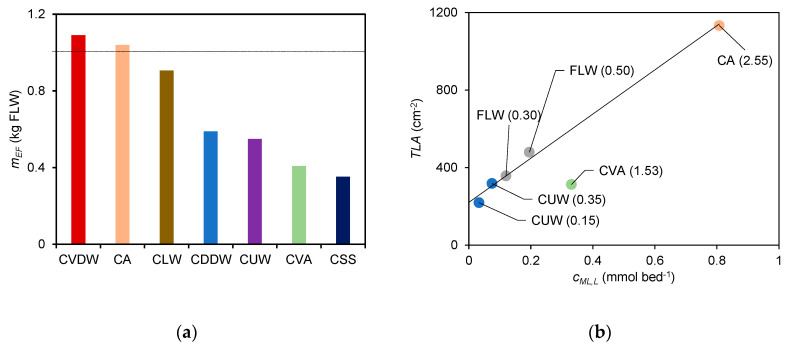
(**a**) Values of the parameter *m_EF_* (equivalent mass of FLW). The horizontal line is settled for *m_EF_* = 1 kg FLW and (**b**) correlation between the total leaf area (*TLA*) of lettuce from a crop field assay and the parameter *c_ML,L_* obtained from metal titrations of the HA extracts.

**Table 1 molecules-28-01503-t001:** Regression analysis parameters of the linear correlations between the total leaf area (*TLA*) of lettuce, and *c_ML,L_* or *c_ML,H_* for the HA, FA and DOM extracts. *TLA* values were obtained from a crop field assay using the composts and the fertiliser whose extracts were characterized by Cd^2+^ titrations. The values from CVA were not included in the correlation analysis.

	Control Assay		HA			FA		DOM
	*TLA* (cm^−2^)	Intercept (cm^−2^)	Slope (cm^−2^ mmol^−1^ bed)	*r**	Intercept (cm^−2^)	Slope (cm^−2^ mmol^−1^ bed)	*r**	*r**
*c_ML,L_* (mmol bed^−1^)	243	221	1139	0.997	210	10,508	0.998	0.051
*c_ML,H_* (mmol bed^−1^)	243	240	348	0.995	193	4220	0.997	0.048

**r*—coefficient of correlation.

## Data Availability

Not applicable.

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
