# Peer review of "Developing a Compost Quality Index (CQI) Based on the Electrochemical Quantification of Cd (HA) Reactivity"

_molecules, 2023, doi:10.3390/molecules28031503_

Round 1
Reviewer 1 Report
The manuscript is well written and carries a novel idea and can be accepted in the current form if. Results figures will be performed in clear form will be better
Author Response
We very much appreciate the Reviewer’s encouraging comments and kind recommendation for accepting our manuscript for publication after minor revision.
- The text was reviewed to improve the reader's comprehension.
Reviewer 2 Report
The work presented a quantification method of compost. The experimental data obtained are sufficient and the explanation seems reasonable. It can be accepted for the publication.
I would suggest providing C/N ratio measurement including the method development and the results can be supplemented.
Author Response
We very much appreciate Reviewer’s kind recommendation for accepting our manuscript for publication after minor revision.
I would suggest providing C/N ratio measurement including the method development and the results can be supplemented.
- The results for the C/N values requested by the Reviewer were obtained in the context of previous studies and can be found in Silva et al., "Sustainability" (2022). The results for the HA and FA extracts can be found in Lopez et al., "JEMA" (2021). Both of these references have been cited in the text. Additionally, a table containing the values from all composts has been included in the supplementary material.
Reviewer 3 Report
The experimental design part of the manuscript is not perfect, and it is difficult to judge whether the experimental method can accurately explain the problem. The logic of writing needs to be further improved. Specific comments are as follows.
1. In the line of 17. When abbreviations appear in the manuscript for the first time, they need to be marked with full names, such as HA, FA and DOM in the abstract (in the line of 17), FTIR、TGA and DSC int the line of 49, and so on.
2. Some important results with data should be included in the abstract.
3. In the lines of 51 to 53. The explanation of this reason seems incorrect. Please check and attach the literature.
4. In the lines of 56 to 60. There are some wrong understanding of soil humus. Such as, humin is a part of the humic substances though it can not dissolved in alkali solution, and fulvic acids can be dissolved in acid, alkali, ethanol and water.
5. The logic of the introduction needs to be sorted out and improved.
6. The writing of the result part should pay attention to English tense.
7. The discussion part lacks strong persuasion.
8. The experimental design part needs to be introduced in detail.
Author Response
The experimental design part of the manuscript is not perfect, and it is difficult to judge whether the experimental method can accurately explain the problem. The logic of writing needs to be further improved. Specific comments are as follows.
- We appreciate the Reviewer's comments and suggestions. We have exhaustively reviewed and considered them, and have made revisions to our manuscript to the best of our ability. The manuscript has been thoroughly reviewed in order to enhance the logical flow of the writing, as suggested by the Reviewer.
1 In the line of 17. When abbreviations appear in the manuscript for the first time, they need to be marked with full names, such as HA, FA and DOM in the abstract (in the line of 17), FTIR、TGA and DSC int the line of 49, and so on.
- In response to the Reviewer's suggestion, full names have been provided for all abbreviations when they are first introduced throughout the manuscript.
- Some important results with data should be included in the abstract.
- In accordance with the Reviewer's recommendation, the abstract has been rewritten to include the most significant results: “The parameter mEF (equivalent mass of fertilizer) was used for this scale sorted the seven composts from 0.353 to 1.09 kg FLW, for CSS (compost of sewage sludge) and CVDW (vermicompost of domestic waste), respectively. The significance of this parameter was verified through a correlation analysis between binding extent and the effect of compost application on lettuce crop growth in a field trial.”
- In the lines of 51 to 53. The explanation of this reason seems incorrect. Please check and attach the literature.
- In response to the Reviewer's request, the text (from line 51 to 53) mentioning the lack of correlation between the growth of a lettuce crop and the abundance of essential nutrients N, P, K has been rewritten to clearly reference the study [3], which presents these results and conclusions: “(…) To address this issue, we conducted a thorough characterization of composts of different origin using various physico-chemical techniques (e.g. elemental analysis, Fourier transform infrared (FTIR), Thermogravimetric analysis (TGA), Differential scanning calorimetry (DSC)) and by a lettuce crop field experiment [3]. The results of the lettuce growth were not correlated with any of the compost physico-chemical parameters, including the essential nutrients like N, P and K. This lack of correlation may be due to the complexity of the compost samples and the presence of a significant number of inert materials, like lignin [3].”
- In the lines of 56 to 60. There are some wrong understanding of soil humus. Such as, humin is a part of the humic substances though it can not dissolved in alkali solution, and fulvic acids can be dissolved in acid, alkali, ethanol and water.
- We are grateful to the Reviewer for highlighting the inadequacy of the term "humic substances" in the context of the aforementioned sentence. We rewrote the sentences in lines 56 to 60 for clarity and replace HS by HA/FA: ”Soil organic matter is extracted with a strong base under a N2 atmosphere, leaving an insoluble fraction named humin. The soluble fraction is further separated by precipitation at pH 1 into the acid insoluble humic acids (HA) and the fully soluble fulvic acids (FA) [4].”
- The logic of the introduction needs to be sorted out and improved.
- We have re-written this section in response to the Reviwer’ suggestion, with the goal of highlighting the two main points addressed in this work: 1) the need of quality parameters for compost that are directly related to its bioactivity, and 2) the properties of humic and fulvic acid extracts that make them good candidates as markers of bioactivity.
- The writing of the result part should pay attention to English tense.
- In accordance with the Reviewer's recommendation, the entire text has been revised to enhance its expression in the English language.
- The discussion part lacks strong persuasion.
- We have taken the Reviewer's suggestion into consideration during the review process and have worked to strengthen the persuasiveness of the text in the discussion of results.
- The experimental design part needs to be introduced in detail
- In this work, tests were conducted using the AGNES voltammetric technique, which may not be familiar to all readers of the journal "Molecules". These experiments followed conventional procedures for this technique. To address the Reviewer's suggestion, a reference with detailed instructions on how to perform AGNES experiments was added to Section 3.4. “Detailed instruction to perform AGNES experiments can be found in the excellent tutorial review by Solis et al. [23] including the experimental steps to implement AGNES, step-by-step laboratory protocols, and two practical case studies. In our case the methodology (…)”
Reviewer 4 Report
Title
Developing a compost quality index (CQI) based on the elec- 2 trochemical quantification of Cd (HA) reactivity.
General comments
The manuscript was exceptionally well written. The approach in this manuscript is very interesting and it will stimulate further debate in the scientific community.
However, I have very few minor comments as below:
Line 44: Is C/N ratio a carbon/nitrogen ratio? If so, the authors have been advised to write it in full since it is mentioned here.
Line 115: Unless I missed it, there is a need to give an explanation why CUW and CLW samples did not follow the trend of the binding capacity of HA>FA>DOM.
Figures 4 and 5: I think the author should consider using different shapes for the data points instead of different colours. Some of the potential readers may be colour-blind.
Author Response
The manuscript was exceptionally well written. The approach in this manuscript is very interesting and it will stimulate further debate in the scientific community.
- We very much appreciate Reviewer’s encouraging comments and kind recommendation for accepting our manuscript for publication after minor revision.
However, I have very few minor comments as below:
Line 44: Is C/N ratio a carbon/nitrogen ratio? If so, the authors have been advised to write it in full since it is mentioned here.
- We thank the Referee’s suggestion. The text was changed to “Carbon/nitrogen (C/N) atomic ratio” as suggested by the Reviewer.
Line 115: Unless I missed it, there is a need to give an explanation why CUW and CLW samples did not follow the trend of the binding capacity of HA>FA>DOM.
- We thank Referee’s comment and helpful suggestion. The text was rewritten (from line 108 to 118) to discuss the possible origin of the general trend and the deviation observed for CUW and CLW extracts.
Figures 4 and 5: I think the author should consider using different shapes for the data points instead of different colours. Some of the potential readers may be colour-blind.
- We thank Reviewer’s attention to the layout of our figures and consideration to colour blind readers. The data symbols were changed and now different shapes are used in Figures 4 and 5.
Round 2
Reviewer 3 Report
None